# VP3 Phage Combined with High Salt Promotes the Lysis of Biofilm-Associated *Vibrio cholerae*

**DOI:** 10.3390/v15081639

**Published:** 2023-07-27

**Authors:** Xu Li, Xiaorui Li, Huayao Zhang, Biao Kan, Fenxia Fan

**Affiliations:** 1School of Light Industry, Beijing Technology and Business University, Beijing 100048, China; 2State Key Laboratory of Infectious Disease Prevention and Control, National Institute for Communicable Disease Control and Prevention, Chinese Center for Disease Control and Prevention, Beijing 102206, China; 3Department of Epidemiology, School of Public Health, Cheeloo College of Medicine, Shandong University, Jinan 250012, China

**Keywords:** *Vibrio cholerae*, biofilm, NaCl, receptor, phage, lysis, biocontrol

## Abstract

Cholera, caused by pathogenic *Vibrio cholerae*, poses a significant public health risk through water and food transmission. Biofilm-associated *V. cholerae* plays a crucial role in seasonal cholera outbreaks as both a reservoir in aquatic environments and a direct source of human infection. Although VP3, a lytic phage, shows promise in eliminating planktonic *V. cholerae* from the aquatic environment, its effectiveness against biofilm-associated *V. cholerae* is limited. To address this limitation, our proposed approach aims to enhance the efficacy of VP3 in eliminating biofilm-associated *V. cholerae* by augmenting the availability of phage receptors on the surface of *Vibrio cholerae*. TolC is a receptor of VP3 and a salt efflux pump present in many bacteria. In this study, we employed NaCl as an enhancer to stimulate TolC expression and observed a significant enhancement of TolC expression in both planktonic and biofilm cells of *V. cholerae*. This enhancement led to improved adsorption of VP3. Importantly, our findings provide strong evidence that high salt concentrations combined with VP3 significantly improve the elimination of biofilm-associated *V. cholerae*. This approach offers a potential strategy to eliminate biofilm-formation bacteria by enhancing phage–host interaction.

## 1. Introduction

*Vibrio cholerae* has two distinct lifestyles: a planktonic form and a sessile form known as biofilms. The biofilms are composed of surface-attached microorganisms surrounded by an extracellular matrix of secreted biopolymers, which provide *V. cholerae* with several advantages to withstand environmental stresses, such as temperature shifts, osmotic stress, bacterivorous predators, and nutrient limitation [1,2,3]. Biofilm-associated *V. cholerae* in natural aquatic environments serves as a reservoir for seasonal cholera outbreaks in Bangladesh [4,5]. It is also the infectious form of cholera, as evidenced by the pathogenicity exhibited in rabbit intestines, a widely used animal model for the study of *V. cholerae* pathogenicity [6]. Additionally, the presence of biofilm-like aggregates in fresh human stool provides a high dose of the pathogen, enhancing infectivity and environmental persistence [7]. Epidemiological studies have shown that removing particles larger than 20 μm from water can reduce cholera incidence [8,9], which highlights the significance of biofilms in cholera transmission. Collectively, these findings strongly support the notion that eliminating biofilm-associated *V. cholerae* from natural aquatic environments could play a crucial role in preventing and controlling cholera transmission.

Phage biocontrol offers a cost-effective and environmentally friendly method for targeting specific pathogenic bacteria [10,11]. In the case of cholera, phage-based approaches have demonstrated promising advancements in both human treatment (during the animal testing phase) and the elimination of pathogenic *V. cholerae* from aquatic environments. Successful results have been demonstrated in animal models, including the effective reduction in *V. cholerae* in the intestines of infant mice through the use of a cocktail of three ICP virulent phages (ICP1, ICP2, and ICP3) [12] and the prophylactic and therapeutic control of cholera in an infant rabbit model using phage Phi_1 alone [13]. Moreover, ICP1 has shown efficacy in preying on *V. cholerae* in estuarine water, indicating its potential in reducing the aquatic reservoirs of *V. cholerae* in epidemic areas [12]. These developments highlight the potential of phage biocontrol as an effective strategy for combating cholera.

Biofilm-associated *V. cholerae*, similar to other biofilm-associated bacteria, exhibits more resistance to lytic phages, posing a significant challenge to their antibacterial effectiveness [14]. Although few studies have focused on eradicating biofilm-associated *V. cholerae*, phages possessing a matrix-degrading enzyme [14], and a combination of antibiotics with phages have shown promise [15]. For other pathogens, various phage-based methods have been explored, including the use of phage cocktails, engineered phages for biofilm mitigation, and phages combined with antimicrobial compounds [16,17].

Cholera, caused by toxigenic *Vibrio cholerae* serogroup O1 or O139, is a major public health issue in developing countries [18,19,20,21]. VP3 is a lytic phage used in the phage-typing scheme for El Tor strains of *V. cholerae* serogroup O1 [22], with potential applications in eradicating *Vibrio cholerae* serogroup O1. However, it has been observed that biofilm-associated *V. cholerae* exhibits a significantly higher resistance to VP3 compared to its planktonic form. To address this limitation, our proposed approach aims to enhance the efficacy of VP3 in eliminating biofilm-associated *V. cholerae* by augmenting the availability of phage receptors on the surface of *Vibrio cholerae.* TolC is a receptor of VP3 [23] and a salt efflux pump present in many bacteria [24,25,26] In this study, we employed NaCl as an enhancer to stimulate TolC expression and observed a significant enhancement in TolC expression in both planktonic and biofilm cells of *V. cholerae*. This enhancement led to improved adsorption of VP3. Importantly, our findings provide strong evidence that high salt concentrations combined with VP3 significantly improve the elimination of biofilm-associated *V. cholerae*. This approach offers a potential strategy to eliminate biofilm-formation bacteria by enhancing phage–host interaction.

## 2. Materials and Methods

### 2.1. Bacterial Strains, Phage, and Culture Conditions

As described in a previous study [25], VP3 phage was propagated on host strain 2477c. A double-layer plaque assay was used to determine phage titers. *V. cholerae* El Tor strain ICDC-VC70 was used for biofilm formation and elimination experiments in this study. An SM-resistant *V. cholerae* strain was used to distinguish from *E. coli* SM10 *λpir* by its resistance to SM. Luria broth (LB) medium, or LB medium plates containing 15 g/liter agar, or soft agar medium, including an additional 0.7% (*w*/*v*) of agar with different NaCl concentrations, were used to grow all the tested strains.

### 2.2. Double-Layer Plaque Assay

As previously described [26], double-layer plaque assays were conducted. The cell cultures (OD600 = 0.3) were mixed with 4 mL of melted 0.7% LB agar at 50 °C and overlaid on nutrient agar plates. Following the solidification of the upper layer, 10 μL of the samples, pre-filtered through 0.22 μM pore size filters (Millipore Corporation, Bedford, MA, USA), was dropped onto the plate and incubated at 37 °C overnight for the detection of phages. The presence of plaques indicates that the strain is sensitive to phages. Plaques were counted to estimate the amount of phage particles in the sample.

### 2.3. Bioluminescence Assays

To test the sensitivity of ICDC-VC70 to phages, pBBR-pmdh-*luxCDABE* reporter plasmid was conjugated into *V. cholerae* strain ICDC-VC70, named ICDC-VC70-lux. The luminescence was measured and normalized against OD600 using a TECAN infinite M200 Pro microplate reader [27]. Bioluminescence assays were performed in triplicate at each time point.

### 2.4. Water Specimen Collection and Preparation

Freshwater (FW) samples were collected from a small pool located in the yard of the National Institute for Communicable Disease Control and Prevention, Beijing, China. After centrifugation at 5000 rpm for 15 min, the supernatant of FW was filtered through 0.22 μM pore filters (Millipore, Shanghai, China) and then sterilized by high-pressure steam at 121 °C for 15 min; 1 mL of the sterilized FW sample was poured onto the LB plate to verify the absence of bacterial growth and *V. cholerae*-related phages by double-layer plaque assay. The sterilized FW specimens were used to prepare liquid suspensions with different NaCl concentrations.

### 2.5. qRT-PCR

In this study, total RNA was extracted from the different cultures of *V. cholerae* strains using an RNeasy kit (Qiagen, Beijing, China). A quantitative real-time reverse transcription PCR (qRT-PCR) was conducted using a One-Step SYBR Primerscript RT-PCR Kit II (TaKaRa, Beijing, China). Relative expression values were calculated as 2^−(Δ*Ct* target − Δ*Ct* reference)^, Ct is the fractional threshold cycle, and the mRNA of *recA* was used as a reference for calculating the relative expression values. The following primer combinations were used: RecA-F (5′-GTGCTGTGGATGTCATCGTTGTTG-3′) and RecA-R (5′-CCACCACTTCTTCGCCTTCTTTGA-3′) for *recA* mRNA; *tolC*-F (5′-GGCACACTAAGCTCTGCCG-3′) and *tolC*-R (5′-CCGTATCTAAGCTTACCC-3′) for the mRNA of *tolC*; VC0229-F (5′-GCGATCACCTTTGTTGTAC-3′) and VC0229-R (5′-GCATAAGGCGTTTTTACCGC-3′) for the mRNA of the VC0229 gene; and VC0231-F (5′-GGTGCCTATGTTGGCCAAG-3′) and VC0231-R (5′-GATGTGCAGTGGTGCATTG-3′) for the mRNA of the VC0231 gene. A control mixture lacking reverse transcriptase was used to ensure that the chromosomal DNA was not contaminated.

### 2.6. Western Blot Analysis

The samples were loaded onto 12% polyacrylamide gels using SDS loading buffer after boiling at 100 °C for 5 min. The sample loading was repeated in the same order three times and run at 180 V for 45 min. Proteins were transferred onto PVDF membranes (Immobilon-P; Millipore, Shanghai, China) following electrophoresis. Using PVDF membranes cut in the middle, two parts were obtained: one part was used to detect CRP using polyclonal antibodies to CRP (Tiangen Biotech, Beijing, China), and anti-TolC polyclonal antibodies (laboratory preparation) were used to detect the membrane marker protein TolC. Anti-mouse and anti-rabbit peroxidase-conjugated AffiniPure IgG (H1L) secondary antibodies (Zhong Shan Jin Qiao, Beijing, China) were used for protein detection.

### 2.7. Phage Adsorption Assay

*V. cholerae* strains were cultured in LB with different salt concentrations for 2 h (OD600 = 0.2) and mixed with an equal volume of phage VP3 (5 × 10^9^ PFU/mL) for 5 min at room temperature. A phage titer remaining in the supernatant was determined after the suspension was centrifuged at 6000 rpm for 10 min. Three duplicates of each assay were performed.

### 2.8. Preparation and Estimation of Biofilms

In borosilicate glass tubes, biofilms were formed as previously described [28]. Briefly, the overnight culture of *V. cholerae* should be diluted 1:100 into fresh medium and inoculated into multiple borosilicate glass tubes at 37 °C for 20 h for biofilm forming. Various final salinities were added to each borosilicate glass tube with biofilm-forming culture and incubated at 37 °C for 1 h; 500 μL of VP3 phage (108 PFU/mL) was added into the culture and was inoculated for 24 h at 37 °C. As described previously [29], residual bacteria in the culture were diluted and counted, and biofilms were observed and stained with crystal violet. To remove the planktonic culture, tubes were gently washed twice with distilled water and fixed with 3 mL of 96% ethanol for 20 min. The ethanol was removed from the tubes, and 2 mL of the 0.5% crystal violet solution was added to each tube and incubated at room temperature for 20 min. After removing the crystal violet, the tubes were gently washed three times with distilled water and dried. To quantify biofilms, 1 mL of dimethyl sulfoxide (DMSO) was used to extract the cell-associated dye, and the optical density of the suspension was measured at 570 nm.

### 2.9. Assay of Biofilm-Dispersing Activity

To identify phages capable of dispersing the biofilms of *V. cholerae*, biofilms prepared in the laboratory were exposed to different phages and examined for degradation of the biofilms and release of free cells. Biofilms of the appropriate *V. cholerae* strains were established on the sides of a series of glass tubes as described above. Free cells were washed away and tubes with biofilms attached to the inner surfaces were retained. The biofilm in one or more representative tubes was measured by staining with crystal violet followed by dye extraction and measurement of OD at 550 nm. The remaining tubes with the biofilms were inoculated with 1 mL LB broth containing 1.5 × 10^7^ PFU/mL of VP3 phage and held at room temperature. Control tubes were also inoculated with 1 mL LB broth but without the phage. The mixed samples from the liquid culture and the biofilm were diluted at different time intervals to count the cell numbers on the plates. Biofilms retained in the tubes after exposure to the phage were also estimated using crystal violet staining, as described above.

In order to identify phages capable of dispersing the biofilms of *V. cholerae*, biofilms prepared in the laboratory were exposed to different phages, and biofilm degradation and the release of free cells were examined.

### 2.10. Statistical Analysis

All experiments were conducted with three biological replicates, and the qRT-PCR experiment consisted of three biological replicates and three technical replicates. The data are presented as means ± standard errors. Statistical analysis was performed using GraphPad Prism 8. Two-tailed unpaired Student’s *t*-test or one-way ANOVA was used to determine statistical significance. * Significant differences by cataract at *p* < 0.05, ** at *p* < 0.01, *** *p* < 0.001.

## 3. Results

### 3.1. Biofilm-Associated V. cholerae Resisted the Lysis of VP3 Phages

We tested the lytic abilities of the virulent phage VP3 with different titers against the biofilms of *V. cholerae* O1 strain. Adding the VP3 phage into borosilicate vials with biofilms attached to the inner surface resulted in a slight biofilm degradation (Figure 1A,B). We tested the change in phage titers representing phage amplification and examined the counts of *V. cholerae* cells located in the biofilm matrix compared to those in the planktonic form. As Figure 1C,D show, when biofilms of *V. cholerae* were exposed to the VP3 phage there was a slight drop in the counts of *V. cholerae* cells of the biofilm matrix or mild elevation in the phage titers in the aqueous phase within 12 h, and no statistically significant difference was observed for biofilm treated with different phage titers (2.5 × 10^6^, 2.5 × 10^8^ PFU/mL). When VP3 was used in *V. cholerae* cells cultured in the planktonic form, there was an increase in phage titer and almost complete elimination of viable cells. This suggests that the phages killed the planktonic cells at a high rate and VP3 phages multiplied in large titers. At a specific detection time, the bacteriophage titer of the planktonic group was tens to hundreds of times higher than that of the biofilm. However, the degrading activity of VP3 lytic phages against the biofilm-associated cells was not very effective.

To understand the resistance of biofilm-associated *V. cholerae* cells to the VP3 phage, we first examined the binding ability of VP3 to the biofilm cells compared with the free cells. We disrupted the biofilms by shaking them with glass beads to disperse the biofilm-associated cells and then mixed them with the VP3 phage. We also mixed planktonic cells with VP3 as a control. After centrifugation, we measured the remaining phage titers in the supernatants. The VP3 infection titers in the supernatants of the planktonic cell group markedly decreased, suggesting effective adsorption to the VP3 particles, whereas the dispersed biofilm cell group had much higher VP3 titers in the supernatants than the planktonic group (Figure 2A). VC0229, VC0231, and *tolC* are known to be involved in the host adsorption of VP3 [21,25,30]. Therefore, we also measured the transcription levels of VC0229, VC0231, and *tolC* in both biofilm-associated and free cells. We found that compared with the transcription levels in free cells, the transcription of *tolC* in biofilm-associated cells was significantly reduced, while the transcription levels of VC0229 or VC0231 showed no difference in both growth states (Figure 2B). Therefore, these results suggest that the lower transcription of the receptor gene *tolC* might play a role in the decreased sensitivity of biofilm cells to VP3 infection.

### 3.2. High Salt Induced the Transcription and Expression of tolC Both in Planktonic and Biofilm Cells

In *V. cholerae*, TolC has multiple functions in bile salt resistance, intestinal colonization, and RTX toxin protein secretion [31] and as a VP3 phage receptor involving the host adsorption of VP3 [21,25]. Salinity is a major water environmental factor affecting bacterial community composition and can shift to some extent [32]. Upregulation of the outer-membrane protein TolC efflux pump was found in salt tolerance regulation in *Yersinia pestis* [22]. In this study, the influence of salinity on the growth of wild-type *V. cholerae* and the *tolC* deletion mutant Vc(Δ*tolC*) was analyzed to test the role of TolC in the salt tolerance; a slight, but not obvious, difference existed in the bacterial density of wild-type *V. cholerae* strains under high NaCl condition during 24 h of incubation, while Vc(Δ*tolC*) had a significant and dose-dependent growth inhibition after *V. cholerae* strains were incubated for 24 h (Figure 3A), especially in the 3% NaCl culture condition (Figure 3A,B). Notably, under the 3% NaCl culture condition, the growth rate of *V. cholerae* strains was found to be faster than that of Vc(Δ*tolC*) strains, starting from the 15th hour, and the difference in OD600 values persisted until after 24 h (Figure 3A,B). After biofilm-associated cells were incubated in LB culture with high NaCl for 2 h, the impacts of high salt on the transcription of *tolC* were further analyzed, and the results showed that high NaCl increases the transcription level of the *tolC* gene (Figure 3C), and the expression level of *tolC* was also greatly improved (Figure 3D). Our data showed that facing high salt pressure, the overall amount of TolC on the bacterial surface increases to play a role in improving the high salt tolerance of *V. cholerae*.

### 3.3. High Salt Promoted VP3 Binding to Biofilm Cells

The quantity and characteristics of the phage receptor, as well as its location on the host surface, can influence phage binding to *V. cholerae*. Phage-binding assays were performed to test whether the high transcription and expression of the gene *tolC* induced by high salt will affect the adsorption of VP3 phages. Dispersed biofilm-associated *V. cholerae* cells, after being incubated with different NaCl concentrations (0.5–3%) for 2 h, were mixed with sufficient VP3 (MOI = 1000:1); the remaining phage titer in the supernatant of each sample was subsequently determined by counting the number of plaque-forming units (PFU) in the supernatant fluid of the centrifuged mixture after a short adsorption period, and the results showed that the VP3 titer in the supernatant of *V. cholerae* strains cultured in the 3% NaCl group was lower than those cultured in the 0.5% and 1.5% NaCl groups (Figure 4A). Fluorescence values were measured in precipitated cells resuspended in SM buffer. Inverse correlations were observed with the fluorescence in the corresponding cell precipitate and the phage titer in the supernatant of each sample (Figure 4B). We further observed VP3 particle adsorption to each biofilm-associated *V. cholerae* cell by confocal laser scanning microscopy (CLSM). SYBR gold-stained VP3 could bind to all these strains, but the fluorescence intensity increased with salt concentrations (Figure 4C).

The adsorption reaction could be influenced by the change of *V. cholerae* strain, as well as by the NaCl tolerance of the phages. To determine whether the high adsorption was due to the change in VP3 phage activity by 3% NaCl, we compared the effect of NaCl on VP3 phage activity at different temperatures (4 °C and 30 °C), and we found that the VP3 phage was stable, and no change in the phage survival was observed by the natural variability in temperature (4 and 30 °C) and salinity (0.5, 1.5, and 3%) after 72 h (Figure 4D). The results indicated that the natural variability in salinity and temperature parameters in aquaculture waters did not significantly affect phage survival. This suggests that the higher phage adsorption of *V. cholerae* in cultures with high NaCl concentration is not caused by a change in VP3 activity, but due to an increased expression of the VP3 phage receptor in *V. cholerae*.

### 3.4. Combined Application of High Salt and VP3 Phage Eliminated The Biofilm of V. cholerae

The effect of salinity on the sensitivity of dispersed biofilm-associated *V. cholerae* to the VP3 phage [21,25,30] was analyzed by double-layer plaque assay and counting the residual *V. cholerae* uncracked by VP3 in a planktonic state. We observed that the formation of phage plaque becomes faster under higher NaCl (1.5, 3% *w*/*v*) concentrations. The rate of *V. cholerae* inactivation in 3% NaCl was faster than those in 0.5% and 1.5% NaCl concentrations, and with the increment in NaCl there was an acceleration of about 5 h before the lytic phenotype of the 0.5% NaCl group was equivalent to that of 3% NaCl broth; this means that the lysis rate of the VP3 phage to dispersed biofilm-associated *V. cholerae* is accelerated with the increase in the NaCl concentration (Figure 5A). Dispersed biofilm-associated cells of *V. cholerae* were added into LB cultures with the NaCl final concentrations of 0.5%, 1.5%, and 3% (*w*/*v*), respectively. The real-time phage therapy kinetics were examined by measuring the residual *V. cholerae* uncracked by VP3 of culture samples from a specific time (0, 4, 8, 12, and 24 h). The rate and efficiency of phage therapy increased with the increase in salt content (Figure 5B).

Due to all the above results being based on *V. cholerae* cells dispersed from biofilm-associated cells or in planktonic states, the inactivation effect of VP3 on *V. cholerae* biofilms was further analyzed to determine whether high salt is conducive to the removal of biofilm-forming *V. cholerae*. The data showed that VP3 significantly decreased the number of bacteria in the biofilm under high salt conditions; the lytic phenomena increased progressively as the salt concentration of the medium was increased; and the destruction of bacteriophage on biofilm formation is more obvious under 3% NaCl conditions (Figure 5C). After being treated with the VP3 phage, the biofilm also decreased significantly under higher salt conditions (Figure 5C,D). The biofilm-associated *V. cholerae* from each group was dispersed and the number of residual *V. cholerae* was counted. The results showed that the concentration of *V. cholerae* was greatly degraded with the improving salinity concentration. Under the condition of 3% NaCl, phages can effectively reduce the number of viable bacteria by 8 logs after 24 h of incubation (Figure 5E).

## 4. Discussion

Phages can penetrate the inner layers of biofilms through water channels [16], making them valuable tools to destroy biofilms by killing bacteria through a lytic life cycle [24,27,28]. The success of phage elimination of biofilm-associated bacteria depends on various factors, including receptor availability, which has been scarcely studied. This study verifies that receptor availability is important for phage-mediated elimination of biofilm-associated bacteria, using TolC and VP3 as examples. The decrease in *tolC* expression may be attributed to the lesser metabolic activity of biofilm-associated cells or the inverse correlation between biofilm formation ability and *tolC* transcription level [29]. The low expression level of *tolC* affects the interaction between VP3 and *V. cholerae*; even if the phage can contact *V. cholerae* in biofilms, it may not support optimum phage adsorption and infection, so the degradation efficiency of the whole biofilm is reduced, as the phage resistance phenotype of biofilms.

Phage-based combination strategies are good antimicrobial therapeutic alternatives for preventing and controlling pathogenic bacteria. The planktonic cells are sensitive to phages, while biofilm-associated cells are mostly resistant to phages; therefore, it is necessary to explore different methods to eliminate the strains in different survival states. The combination approach can be established from different points, such as phage and phage-derived enzyme combination to inhibit the formation of pathogenic bacterial biofilm and a high lysis effect on existing biofilms; the use of phage cocktails based on expanding host range coverage; phage conjunction with antimicrobial drugs displaying a tremendous synergistic effect on bacterial mortality. Finding a factor that can improve the ability of phages to eliminate biofilms can expand the scope of phage applications. It is a new strategy to prevent and eradicate infectious bacterial biofilms by increasing the expression of phage receptors. Our study established a new combination method from this new perspective: the combination of the VP3 phage with salinity that can promote the expression of the phage receptor of VP3 to increase the adsorption capacity of the phage to *V. cholerae* and accelerate cell lysis. In clinical settings, combined treatments such as multiple antibiotics with a single phage [30] or a phage cocktail with a single antibiotic [31], have been found to be more effective in eradicating biofilms due to the multiple mechanisms involved. Therefore, the compounds identified by our strategy may be used in conjunction with existing combined treatments to further enhance efficacy against target pathogens.

High salt mainly affects *V. cholerae* rather than the VP3 phage, making *V. cholerae* more sensitive to another new pressure. Although different free phages differ in their NaCl tolerance [32], the VP3 phage itself is salt- and temperature-stable (Figure 4D). VP3 phage survival was maintained relatively constant when the phage was tested at various salinity and temperature parameters. Salinity has a significant effect on bacterial protein expression, especially those associated with salt tolerance; the higher expression of TolC of *V. alginolyticus* [25], and the activation of TolC efflux pump overexpressed in *Yersinia pestis* [24] and *Salmonella typhimurium* [26] in NaCl osmotic environment stresses were observed. Although high salinity slightly inhibits the growth of *V. cholerae*, it is conducive to phage lysis. The main function of NaCl is to improve the transcription and expression of the *tolC* gene encoding the outer membrane protein TolC, which has a dual identity as a salt efflux pump and a phage receptor. This improvement in high salt tolerance will eventually lead to higher opportunities for VP3 phage adsorption and affinity, resulting in faster bacteria clearance.

Research on the effect of the salinity content on the efficiency of phage therapy for the disinfection of biofilm-associated *V. cholerae* is rarely reported. Various physical and chemical (water temperature, salinity) conditions can influence the efficiency of phage therapy [33]. The effects of these factors on phage survival and activity are different [34]. Salinity is a major determinant that shapes bacterial communities over seasonal and time changes [35,36,37]; the effects of salinity on bacterial diversity vary, including a negative correlation [36,38,39], no effect [40], or a peak diversity at low salinity [41]. The ability to tolerate salt is important for bacteria to survive and thrive in severe environments, so salinity is a ubiquitous and relatively mild candidate factor for phage combination therapy. *V. cholerae* can grow in ocean–river intersections with low salinity and marine environments with high salinity. In addition to the different salinity requirements for the growth of different kinds of bacteria, salinity also influences the interaction between bacteriophages and their hosts and ultimately affects the ecological composition of bacteria.

This study sheds light on the molecular mechanism behind the effect of salinity on phage therapy by demonstrating that salinity regulates the transcription and expression of the VP3 bacteriophage receptor TolC. Based on this, NaCl–VP3 therapy was established as a new approach to combat biofilm-associated *V. cholerae*. The successful infection processes, such as phages of *V. anguillarum*, and *V. vulnificus* [42], require a specific salinity range. Phage lysis of Streptococcus lactis was stimulated by NaCl [43], and the adsorption of phage could be influenced by varying the electrolyte concentration [44]. Thus, salinity is the dominating environmental factor controlling bacterioplankton microbial communities in freshwater lake systems or saline habitats [45,46,47,48]; however, little is known about the mechanism. Our study shows that the salt concentration of the medium can affect the phage–cell relationship by modulating the expression level of TolC in *V. cholerae*. The limitations of the NaCl–VP3 combination therapy proposed in this study were only validated under controlled laboratory conditions. Therefore, in order to discuss its potential application in seafood preservation and sewage systems, further experimental validation is necessary.

We propose that the NaCl–VP3 combinational therapy shows promising potential for both seafood preservation and sewage systems. Fish can act as hosts for *V. cholerae*, which can be found in various organs, such as the intestinal tract, scales, and gills [49], thereby leading to contamination during the processing and transportation of aquatic products. The formation of biofilms by *V. cholerae* strains enhances their resistance to disinfectants, particularly in developing countries with limited disinfection practices, thereby increasing the risk of transmission. To address these challenges, the NaCl–VP3 combination therapy can be developed as a phage product for disinfecting seafood containers. This therapy can be applied through soaking or spraying methods, effectively eliminating biofilms from the corners and interiors of the containers. However, further optimization of preservation methods for these phage products is necessary to ensure their practical application. Moreover, the NaCl–VP3 therapy holds the potential for addressing sewage systems in areas where cholera is endemic. During cholera outbreaks, inadequate sanitation facilities directly contribute to the contamination of water with biofilm-like aggregates from infected individuals, thereby facilitating the rapid spread of the infection [50]. Although membrane filtration has been the prevailing method for sewage treatment in recent decades, biofilms continuously reduce the flow rate of wastewater through the membrane surface, significantly impairing its efficiency. Phages have been proven effective in mitigating membrane fouling [51]. Therefore, integrating NaCl–VP3 therapy with membrane filtration may offer a potential solution for effectively eliminating *V. cholerae* from sewage. Furthermore, the modification of the VP3 phage genome by incorporating genes encoding matrix-degrading enzymes, or the utilization of phage cocktails instead of a single VP3 phage, can further enhance the eradication of biofilm-forming bacteria. This represents an extension of the current study, offering opportunities for future research and development in this field.

## Figures and Tables

**Figure 1 viruses-15-01639-f001:**
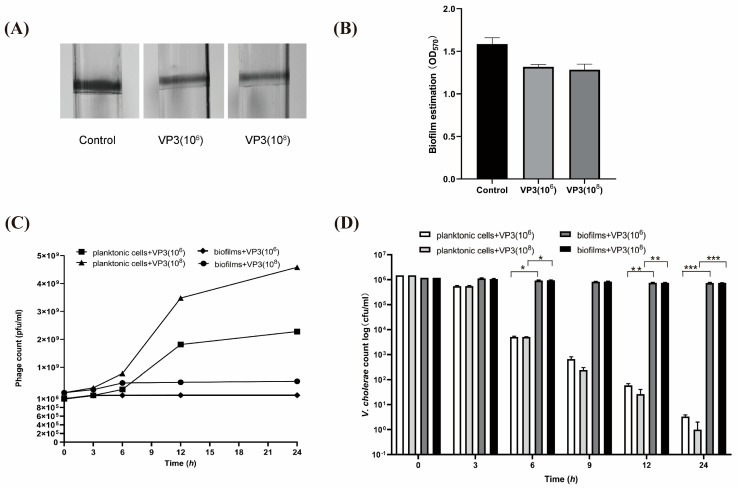
Analysis of phage susceptibility of planktonic and biofilm-associated cells to the VP3 phage. Biofilms of *V. cholerae* were established on the sides of glass tubes by static incubation in an LB medium. After washing away planktonic cells, the VP3 phages with different titers (2.5 × 10^6^, 2.5 × 10^8^ PFU/mL) were incubated with the biofilm for 12 h, the representative degradation effect was measured by staining with crystal violet (**A**), and the crystal violet staining biofilm result was quantified at 570 nm (**B**). Initial fresh planktonic cell cultures (1.5 × 10^6^ CFU/mL) were mixed with VP3 to 10^6^ and 10^8^ PFU/mL final phage titers, respectively. Moreover, the biofilm-associated cells (1.2 × 10^6^ CFU/mL) after washing away planktonic cells were incubated with VP3 phages (10^6^ and 10^8^ PFU/mL, respectively). Real-time phage titers in the aqueous phase and the number of viable *V. cholerae* cells from biofilm were determined at the indicated times (**C**,**D**). In the above, asterisks mark statistically significant differences (* *p* < 0.05, ** *p* < 0.01, *** *p* < 0.001).3.2. VP3 Adsorption and the Receptor Gene tolC Transcription were Decreased in the Biofilm-Associated Cells.

**Figure 2 viruses-15-01639-f002:**
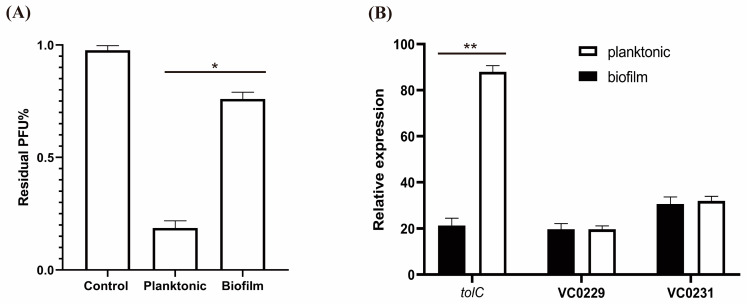
Phage adsorption ability and *tolC* transcription levels in the planktonic and biofilm-associated cells. (**A**) Analysis of VP3 adsorption to planktonic and biofilm-associated cells. LB culture medium containing only the VP3 phage was used as a control, and the phage titer in the control supernatant was set to 100%. Error bars, s.d. (n = 3 biological replicates), two-tailed unpaired Student’s *t*-test. (**B**) qRT-PCR assays of genes VC0229, VC0231, and tolC transcription in the planktonic and biofilm cells. Error bars, s.d. (n = 3 biological replicates with 3 technical replicates), one-way ANOVA test. In the above, asterisks mark statistically significant differences (* *p* < 0.05, ** *p* < 0.01). The absence of asterisks indicates that there is no statistical significance.

**Figure 3 viruses-15-01639-f003:**
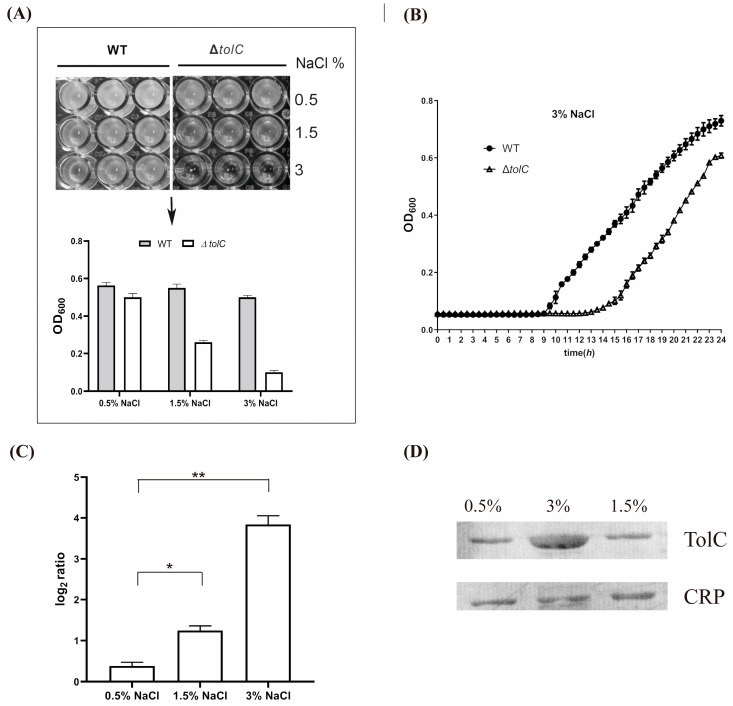
Transcription and expression of *tolC* and the growth of *V. cholerae* in different salt conditions. (**A**) The growth of *V. cholerae* and its *tolC* gene mutant in different salt concentrations; upper: observation of the effect of salt on the growth of the wild-type strain and VC(Δ*tolC*); lower: analysis of the cell concentration by testing the OD600 values. Wild-type (WT) strain and VC(Δ*tolC*) were incubated in different salt cultures for 8 h, and the growth of the strain was observed. Error bars, s.d. (n = 3 biological replicates). (**B**) The real-time growth of the wild-type strain and VC(Δ*tolC*) was detected in a 3% NaCl culture for 24 h. Error bars, s.d. (n = 3 biological replicates). (**C**) qRT-PCR assays of *tolC* transcriptions under different salinity. Error bars, s.d. (n = 3 biological replicates with 3 technical replicates), two-tailed unpaired Student’s *t*-test. (**D**) Immunoblot analysis of TolC in cell fractions of *V. cholerae* strains in different salt conditions. Samples were subjected to SDS-PAGE and immunoblot analysis using anti-TolC polyclonal antibodies (laboratory preparation) or CRP polyclonal antibodies (Tiangen Biotech, Beijing, China). In the above, asterisks mark statistically significant differences (* *p* < 0.05, ** *p* < 0.01). The absence of asterisks indicates that there is no statistical significance.

**Figure 4 viruses-15-01639-f004:**
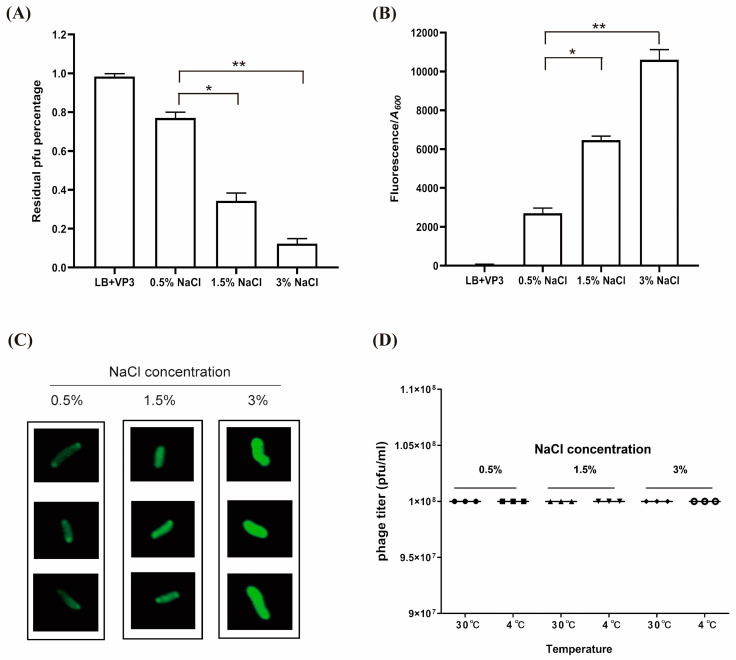
Detection of VP3 adsorption of *V. cholerae* cultured in different salt concentrations. (**A**) VP3 adsorption assays of V. cholerae cultured in different salt conditions. *V. cholerae* strains were mixed with phage (5 × 10^9^ PFU/mL) according to MOI = 1000; the concentration of *V. cholerae* is far lower than phage titer to fully measure the adsorption capacity of a single *V. cholerae* cell to VP3 phage and the adsorption ability was determined by residual phage titer in the supernatant of each sample. Error bars, s.d. (n = 3 biological replicates), two-tailed unpaired Student’s *t*-test. (**B**) Binding capacity between the strain and VP3 is measured via total fluorescence value/A600. LB culture medium containing only VP3 phage was used as a control, and the phage titer in the control supernatant was set to 100%. Error bars, s.d. (n = 3 biological replicates), two-tailed unpaired Student’s *t*-test. (**C**) Observation of a single *V. cholerae* cell with SYBR gold-labeled VP3 binding on the surface by CLSM. (**D**) Detection of the effect of temperature and NaCl on the activity of the VP3 phage. In the above, asterisks mark statistically significant differences (* *p* < 0.05, ** *p* < 0.01). The absence of asterisks indicates that there is no statistical significance.

**Figure 5 viruses-15-01639-f005:**
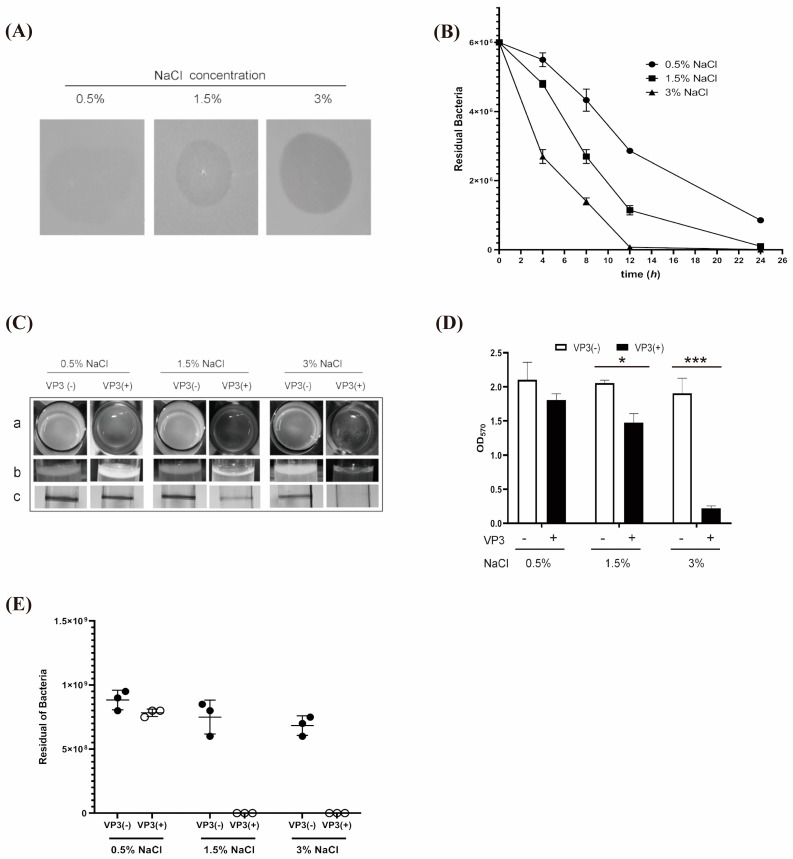
Test of VP3 elimination efficiency of biofilm-forming *V. cholerae* under high salt concentrations. (**A**) Detection of VP3 infection in *V. cholerae* by a double-layer plaque with different salinity concentrations. The wild-type strain ICDC-VC70 (WT, sensitive to VP3) was used and the plaque formation was observed after incubation for 8 h. (**B**) Analysis of VP3 infection in *V. cholerae* in a planktonic state with different NaCl concentrations. *V. cholerae* strains were mixed with VP3 (MOI = 10) in cultures with different salt concentrations, and the residual *V. cholerae* was measured by values of culture samples from a specific time (0, 4, 8, 12, and 24 h). Error bars, s.d. (n = 3 biological replicates). (**C**) Result for the salt–VP3 phage combination method on the biofilm-forming *V. cholerae*. *V. cholerae* strain was cultured to form a biofilm, then different NaCl concentrations were added for one hour of incubation, and the biofilms were finally treated with the VP3 phage: VP3(+); without VP3 as control: VP3(−); the biofilm-forming was observed; each column data represents the observation results of different views (a: vertical observation; b: horizontal observation) or the view by crystal violet staining. (**D**) Quantifying the crystal violet staining biofilm at 570 nm. Error bars, s.d. (n = 3 biological replicates), two-tailed unpaired Student’s *t*-test. (**E**) Cell count of the residual *V. cholerae*. The mixed samples from the liquid culture and the biofilm were diluted to count the cell numbers on the plates. In the above, asterisks mark statistically significant differences (* *p* < 0.05, *** *p* < 0.001). The absence of asterisks indicates that there is no statistical significance.

## Data Availability

All data relevant to this study are included in the article.

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
