# Peer review of "VP3 Phage Combined with High Salt Promotes the Lysis of Biofilm-Associated Vibrio cholerae"

_viruses, 2023, doi:10.3390/v15081639_

Round 1

Reviewer 1 Report

Li et al. have found that high salt increases phage efficacy against biofilm-forming Vibrio cholera by enhancing TolC expression and adsorption of vibrio phages. The study is well designed and well-written, and I don't have any major comments on their work. However, I suggest that the authors share their thoughts, in the discussion, on how their suggestive combinational therapy could be applied and its potential administration route.

Author Response

Please refer to the attached document for our response to the reviewer's comments.

Reviewer 2 Report

The aim of this interesting study was to evaluate the VP3 phage combined with high salt in the lysis of biofilm-associated Vibrio cholerae. However, here are my considerations for improving the manuscript:

- Add the objective of the study in the abstract and at the end of the manuscript introduction.

- Add a study conclusion in the abstract and at the end of the manuscript with the same central idea.

- Add in the penultimate and last paragraph of the manuscript discussion, the limitations of the study, as well as future perspectives.

- The most current reference in the study is from the year 2020. Therefore, check that all references are essential for the introduction and discussion of the manuscript and add or replace older references with references from 2021 to 2023.

Author Response

(The authors gave the same response as above.)
